# Fatigue Crack Growth and Fracture of Internal Fixation Materials in In Vivo Environments—A Review

**DOI:** 10.3390/ma14010176

**Published:** 2021-01-01

**Authors:** Kailun Wu, Bin Li, Jiong Jiong Guo

**Affiliations:** 1Department of Orthopedics, The First Affiliated Hospital of Soochow University, Suzhou 215000, China; Wukailun666666@163.com (K.W.); binli@suda.edu.cn (B.L.); 2Department of Orthopedics, Suzhou Dushuhu Public Hospital (Dushuhu Public Hospital Affiliated to Soochow University), Suzhou 215000, China; 3Orthopedic Research Unit, Soochow University, Suzhou 215006, China

**Keywords:** fatigue crack growth, fracture, internal fixation, alloys, absorbable materials, design flaw, elastic modulus, compatibility, allergic reaction

## Abstract

The development of crack patterns is a serious problem affecting the durability of orthopedic implants and the prognosis of patients. This issue has gained considerable attention in the medical community in recent years. This literature focuses on the five primary aspects relevant to the evaluation of the surface cracking patterns, i.e., inappropriate use, design flaws, inconsistent elastic modulus, allergic reaction, poor compatibility, and anti-corrosiveness. The hope is that increased understanding will open doors to optimize fabrication for biomedical applications. The latest technological issues and potential capabilities of implants that combine absorbable materials and shape memory alloys are also discussed. This article will act as a roadmap to be employed in the realm of orthopedic. Fatigue crack growth and the challenges associated with materials must be recognized to help make new implant technologies viable for wider clinical adoption. This review presents a summary of recent findings on the fatigue mechanisms and fracture of implant in the initial period after surgery. We propose solutions to common problems. The recognition of essential complications and technical problems related to various approaches and material choices while satisfying clinical requirements is crucial. Additional investigation will be needed to surmount these challenges and reduce the likelihood of fatigue crack growth after implantation.

## 1. Introduction

Fatigue crack growth and fracture of orthopedic implants are commonly attributed to the progressive damage under the condition of periodic and intermittent stress load [1]. A sharp increase in the number and size of the micro-cracks is associated with the alternating stress and magnitude of force. When multi-cyclic compression reaches the ultimate tensile strength and as progressive damage accumulates, the internal fixation eventually breaks [2]. The median fatigue limit (MLF) is defined as the minimum load that results in implant failure if exposed to 42,000 cycles, equivalent to about 8 days of walking, on the base of a transverse fracture at the midshaft of femur [3,4]. However, the growth of cracks is a very complicated process in in vivo environments, which are strongly influenced by the biomechanical changes as well as inflammatory response after implantation [5].

Implant failure is usually caused by a network of cracks in the fixation that join and intersect. This network is also referred to as a cracking pattern or as river marks, similar to the branches and small tributaries of a river on a floodplain, as depicted in Figure 1 [6]. River patterns show the direction of fatigue crack progression and are most frequently seen in the relatively fast growing sections of the fatigue zone [7]. The strong correlation between river marks and functional properties of implant allows one to estimate the degree of material degradation. Following Griffith’s seminal work [8] concerning a model predicting the crack initiation, subsequent investigators added to the understanding of plasticity [9], toughness [10], and complex geometries [11,12,13]. In orthopedic, fracture mechanics has been exploited to study microstructural damage and energy at the implant-bone interface [14,15]. An important parameter to describe the morphology of the river marks is notch sensitivity, which is given by the fatigue notch factor (Kf) [16]. Kf is the ratio of the fatigue strength of a smooth specimen to that of a notched specimen [17]. The literature provides almost no evidence that the notch sensitivity correlates with the structural integrity of the implant.

In terms of biomechanical behavior, the potential hazards of the surrounding soft tissue, blood supply, and severe bone defect play a critical role in the propagation and development of cracks, and their importance has been demonstrated by numerous studies [18,19]. During surgery, the dissection of periosteum and surrounding tissue is inevitable in pursuit of strict fixation and anatomical reduction, which can substantially violate the principle of biological fixation and lead to delayed union and nonunion [19]. Under these circumstances, instability at the fracture site will greatly increase the load acting on the internal fixation. The tension on the implanted device can also be created by repeated bending stress. The duration of excessive load and impact, meanwhile, will increase the longer it takes for the bone fracture to heal [20].

The analysis of crack growth draws on many fields of science, including the biomechanical effect, fracture mechanics, image analysis, and materials engineering [21]. Despite advances in theory and surgical techniques, the risk mitigation of fixation failure resulting from biomechanical effects is still in its early stages. At least, technical solutions are reaching a crossroad on this point. Consequently, the latest publication on the applicability of materials has been thrust into the limelight as we strive to lower the incidence of fatigue crack growth [22,23,24]. Currently, stainless steel, titanium (Ti), and cobalt-chromium (Co-Cr) alloy are the major materials used for the fabrication of the orthopedic inner fixing apparatus [25]. In international clinical practice, the most widely used medical stainless steel materials for surgical implant are austenitic stainless steel-316L, 317L, etc. [1]. Starting with austenitic stainless steel, titanium is added to make the material more resistant to corrosion, increase fatigue strength, lower infection risk, and improve the isoelasticity of bones, Ti–6Al–4V is the most frequently used titanium alloy for biomedical application [26]. Hardness is principally improved by adding molybdenum and through the reduction of sulfur, phosphorus, and other impurities [27,28]. The dominant properties of nickel-free nitrogen-containing stainless steel are rust resistance and toughness, which make the materials adapt better to the biochemical environment in vivo [29].

However, these alloys can suffer drastic loss of fatigue endurance through severe notch sensitivity effect caused by notches or stress raisers [30,31]. In addition, these metallic biomaterials can release the toxic metal ions and particles throughout the process of corrosion or wear, which will contribute to inflammatory cascade reaction, changes of biocompatibility, osteolysis along the implant tracks, and loss of fixation [32]. In addition, the elastic modulus of current materials does not provide a good match with natural bone tissue. This reinforces the stress shielding effect from plate and interferes with the formation and reconstruction of new bone [33]. Ultimately, any instability of the implant in the body will have to be solved by a second operation. This increases the extra sufferings and medical expenses of patients.

Given the gravity of this possibility, it is essential to determine specific factors that contribute to and are associated with fatigue crack growth. In the present paper, we discuss state-of-the-art knowledge on fatigue crack growth and fracture modes of orthopedic implants. The work is both a review of the rationale of fatigue crack growth and the state of knowledge about material properties in vivo. We also focus on the properties of materials. Our aim is to examine a number of possible materials innovations for orthopedic, and to provide insights useful in clinical practice and implant design that may speed up the development of new products.

## 2. Inappropriate Use of Internal Fixation

### 2.1. Reshaping of Internal Fixation


Currently, internal fixation still does not match the anatomy of human skeleton perfectly [34]. To remedy the situation, plates are often bent repeatedly to fit snugly onto the bone surface during the aesthetic procedure. Unfortunately, this reshaping can greatly decrease the strength of the plate and speed up the erosion of metal, a primary reason for cracking. At this point, pores close to the surface will become nuclei for fatigue cracks [35]. Subsequently, a sudden release of elastic energy effortlessly can cause the local tensile strength of the nail to be exceeded, resulting in acute failure. Indeed, in plate bending, although the compressed threads tend to disperse minimal stress concentration effect, hole threads still can act as notches and diminish the fatigue strength even with minimal stress intensity, especially in titanium devices [6]. The fatigue strength may also be impacted by bending frequencies, displaying an obviously positive correlation [36]. It has been reported that the fatigue resistance of bending could be enhanced by exerting a sufficient compressive stress on the implant surface and reducing the crystallite size, which has been realized through sandblasting [37]. According to the operation specification, it is strictly forbidden to bend the site of a screw hole, in order to maintain the mechanical strength of steel counterpart. Slight bending between threaded holes is permitted if necessary. Fortunately, these configurational incongruities may be perfectly tackled by customization via 3D printing to enhance implants’ survivorship of fracture fixation in the short run [38].

### 2.2. Interaction of the Different Materials

A sandwich of two different metals can induce a continuous micro-current (a.k.a galvanic corrosion) [39]. A similar situation arises with internal fixation in vivo. In clinical practice, the fixations made from different material compositions may have to be selected in view of a lack of matching apparatus or using stainless wire to tie up bone fragments. Despite the fact that a stable oxide film between the high-quality metals is expected to suppress galvanic corrosion in theory, it was still reported in the literature that a micro-current between the plate and screws can cause electrolytic corrosion and substantially decrease the strength of fixation [40,41].

Besides, in the case of the implants with unstable fixation, scratch and friction damage of the oxide film have been detected at the interface, which could explain why galvanic corrosion occurred [42]. Depending on the electrolyte solution in vivo, the passive oxide film may undergo a cycle of the dissociation and re-oxidation, where the absence of the dissolved oxygen dramatically hampers the repair of the oxide film. Regarding stainless steel, electrolytic corrosion often appears as pit corrosion due to the perforation from chlorine ions, or crevice corrosion as a result of the inhomogeneous distribution of the dissolved oxygen [43]. The primary factors influencing the strength of the galvanic coupling are not only the corrosion potential difference caused by composition of each constituent phase but also the effective contact area of Cathode/Anode. Therefore, weak galvanic corrosion may take place even at the interface between the identical alloys, at least in theory, attributed to minor differentials in the impurity distribution. Fatigue crack initiation can take place at the base of galvanic corrosion. Thereafter, the crack initiation shifts towards the shoulder and the mouth of the pit as the depth of the corroded site increases [44].

### 2.3. The Weakest Link of Internal Fixation

It is extensively known that the hole structure is the weakest link of the fixation [31]. If the weakest links are placed in the vicinity of a fracture, the local stress will be concentrated on the transitional sites, which may be capable of destroying the structure of the plate near the crack limit [4]. In short, an implant is only as strong as its weakest link. As an evidence of the propagation of fatigue cracks in a locking compression plate (LCP), the striations or fatigue cracks are seen first to initiate from a subsurface inclusion embedded under the surface of compression hole and the surface of the locking hole. Once cracks are initiated, the residual stress and microstructure affect the fatigue crack growth [45]. Then both cracks propagate inside the plate [4]. Moreover, it has also been observed that the thread crest is subjected to a maximal stress under bending load and has a tendency to be the fatigue crack initiation site [6]. Kanchanomai et al. found that the propagation of cracks from the initiation site to the bottom part of plate required nearly 5000 cycles in fatigue tests [4]. It is therefore necessary to choose a plate with even numbered holes or without center holes, so as to avoid placing the middle hole on top of the fracture site.

According to the work by Lin et al., the modification of screw hole structures can improve the fatigue strength of plate effectively by removing the threads at the tension side and increasing the crest radii of the threads (Figure 2) [6]. It was also reported that locking buttons were utilized to plug the empty holes adjacent to a defect, that could be implemented to increase the fatigue strength and fatigue life by 4 times compared with plates with unfilled holes (Figure 3) [46,47,48]. The usage of two buttons for locking raised the survival probability. However, the authors did not to date test the strength of the construct adjacent to the fracture site for using a single button or in Long-span comminuted fracture. Further studies, especially including the fatigue crack propagation of the LCP engaged by locking buttons, will be necessary to better predict the fatigue life of an implant in a given clinical situation.

## 3. Design Flaws of Internal Fixation

Design flaws are a key reason for fatigue crack growth in internal fixation [49]. Take the case of intramedullary nail. The design of tubular intramedullary nailing cannot provide a large amount of transverse elastic space depending on the need for weight-bearing exercise. Accordingly, the nail has to be placed under low resistance in the practice of medullary nail due to its transverse fragility [50]. Moreover, broken interlocking screws can be found more frequently if small-diameter nails are used [51]. Semeer and colleagues speculated that vibration of such slightly loose nails in the canal may corrode the bolt at the nail-bolt interface and ultimately lead to further bolt weakening and the emergence of cracks [52]. Therefore, we recommend a nail, which is 1.0 to 1.5 mm smaller than the ultimate size of intramedullary reaming in practice. Otherwise, the split of diaphysis and deformation of the intramedullary nail may easily commence when the nail is hammered into the coarser medullary cavity by force [53].

For the treatment of fracture, blood supply and stability are prerequisites for fracture union, and the axial interfragmentary motorization can expedite the course of bone remodeling [54,55]. However, untimely dynamization gives rise to a range of disastrous complications, such as limb shortening, deformity of rotation, and nonunion [56]. In order to take this into account, early dynamization is recommended, unless the fracture still manifests no signs of healing three months after treatment with interlocking intramedullary nails or a comminuted fracture is identified [57]. With regard to unstable fracture configuration, internal fixation could not ensure stability with low anti-torsion capacity and poor shear strength on the vertical axis after being changed into dynamic fixation, in spite of compressing the fracture end. The main factors contributing to this phenomenon incorporate the traction of muscle and resorption of the fracture end [57]. To cope with this, Dailey et al. have proposed a novel intramedullary nail design, which can generate a stimulatory micro-motion under minimal weight-bearing loads on the strength of cadaver observation (Figure 4) [54]. In the future, an analogous design may secure a dominant position on medical-instrumentation platforms.

In the case of middle and distal tibia fracture, the intramedullary nail ought to be implanted into the distal side of cancellous bone below the isthmus. However, the distal portion of intramedullary nail is usually difficult to anchor depending on open cancellous bone [50]. It means that the tip of intramedullary nail is easy to move around or impact medial wall of bone cortex, and this eventually causes a fatigue crack or break in the implant. Hence, this imperfection of design has to be overcome by using blocking screws, which in a sense can limit the wiggle of the tip by guaranteeing the length of the nail [58].

## 4. Elastic Modulus of Internal Fixation

The strength of fracture healing is, as a rule, supposed to manifest a negative correlation with the elastic Young’s modulus of the implant. The stiffness of metallic implant is an order of magnitude greater than that of cortical bone in general [59]. For instance, the elastic modulus of Ti–6Al–4V (TAV) (124 GPa) far outstrips cancellous bone (3 GPa) or compact bone (12–17 GPa) [60]. This mismatch of the elastic modulus between the implant and bone tissue is a frequent contributor to fatigue crack growth, which often results in the poor load transfer from the implant to the adjacent bone tissue. In the light of the tests carried out so far, stress shielding effects that follow can generate the resorption of bone around the implant, and eventually induce the loosening or breakage of fixation, especially for metal plate with high elastic modulus [61]. A diminution of the elastic modulus of the implant could abate stress concentration in the cancellous and cortical bone by 15% and 16%, respectively [62]. In addition, an implant with a modulus closer to the surrounding trabecular bone could induce a more even repartition of stress and less micro-motion at the interface with the bony bed [63].

Compared with normal bone tissue, the moduli mismatch is more serious in osteoporosis with anticipated high rates of implant crack growth and failure. Thus, some novel implants with ultra-low elastic modulus like Ti-24Nb-4Zr-7.9Sn (42 GPa) [64], Ti-24Nb-4Zr-8Sn (49 GPa) [65], and Ti-50.7at%Ni (29 GPa) [66] were invented for orthopedic applications by biomaterial scientists, which were able to relieve the “stress shielding effect” caused by the modulus mismatch and suppress bone resorption for long term implantation. Mazigi et al. concluded that the Ti-35Nb-3Zr alloy (85 GPa) with a direct biocompatibility test exhibited a greater potential for long-term successful performance compared to the TAV [67]. Ideally, topological design and an appreciation for implant metallurgy should give priority to rigidity at the early stage of fracture healing, whereas flexibility is more important at the later stage. Hence, for the time being, mainstream opinion considers that the flexibility and rigidity of internal fixation are of equal importance.

In the near future, it seems likely that patented technologies, combining an absorbable exterior metal and a flexible interior material, could come to a future implant as a function of the elastic properties of the construct. For such a device, this elastic artefact needs not to be removed, even if it is non-absorbable. On the one hand, the promising custom-made implant has little effect on recipients and lowers the incidence of re-fracture considerably owing to the favorable stability and flexibility. In particular, interior fixation can even become an auxiliary navigator at time of re-fracture if applied in PFNA-II (proximal femoral nail antirotation-II) or Gamma-3 (third-generation gamma nail). On the other hand, the exterior can facilitate the process of fracture healing theoretically by releasing metal ions, such as magnesium and titanium ion.

## 5. Biological Compatibility and Corrosiveness of Implants

Sustaining multiaxial loading, including tension, compression, bending, and torsion, is one prime challenge of implants which also must survive in a very corrosive medium in vivo (high concentrations of enzymes, proteins, salts) [68]. Subject to this cooperative attack, load-bearing implants are prone to corrosion and some inevitable level of wear and tear [69]. It is a fact that corrosion resistance is adequate even in the presence of cracks, but not necessarily once a dynamic load is exerted. This can bring about the irreversible breakage of the passivation layer so that the metal ions spread towards the surrounding tissue.

Compound interactions, like ion exchange or adsorption of proteins, determine the quality and stability of the bone-implant-interface [70]. These redox-reactions may cause conformational variations of biological macromolecules transforming native proteins into antigens which notify the immunological system to recognize an artificial implant as a foreign body [71]. Besides, the surface of the implant can become a fascinating battleground for the spontaneous degeneration and infiltration with inflammatory cells [72,73]. The degradation products are in turn liable to incite aseptic inflammation. The former can produce toxic side-effects, while the later can lead to a total loss of material cohesion [74]. Some relevant studies have conducted a deep analysis of the stress intensity threshold for fatigue (Kmax,th) in corrosive media [75]. The results indicated that corrosive environment possesses a time-dependent attribute, contributing to fatigue crack growth even when stress intensity factor (Kmax) is less than stress intensity threshold for stress corrosion cracking (KIscc) (Figure 5).

Generally, the surface will experience four different corrosion phases, namely immunity, active metal dissolution, passivity, and transpassive metal dissolution [76]. The surface is defined as the electrochemical context where a strongly adherent surface oxide film of a 2–10 nm thickness is present [77]. On the one hand, such an oxide film reduces the dissolution rate of the metal by acting as a physical barrier limiting the transport of electrons, cations and anions between the metal and the electrolyte, and reducing the kinetics of the anodic and cathodic reactions underlying the corrosion process. On the other hand, the deposition of oxide or nitride particles could induce the adherence of tissue to the rough surface of implant, as depicted in Figure 6 [78]. The thickness of the passive film on alloys and its composition changes with potential [77]. The actual value of the corrosion potential is determined by the relative velocities of the formation of novel surface modification and its repassivation [79]. The velocity of the repassivation itself depends on the kinetics of the metal surface reactions and the conditions for oxygen access. However, the repassivation of a mechanically impaired surface areas is hardly possible in the oxygen-deficient medium so any fatigue crack advances faster.

Titanium and its alloys form a very stable oxide layer in quasi-physiological environments bestowing them with exceptional biocompatibility as compared to other metal implant materials [80]. Pitting and crevice corrosion have scarcely been found on implants of Ti-alloys up until now [81]. Their interfacial reaction products predominantly consist of anatase and rutile (TiO2) represented by a tetragonal lattice structure [71]. In contrast to the ion-conducting passive films formed on stainless steel or cobalt and nickel-based alloys, Ti forms a semi-conducting passive layer. A further property of the layer is the regeneration of the oxide layer in milliseconds even after damage in poorly oxygenated media. Depending on the phases that can be retained at room temperature, Ti alloys are classified into three categories: α, α+β, and metastable β alloys [82]. According to a recent investigation, corrosion attack in α/β Ti alloys often initiates at the α/β interfaces, as multi-step process [83]. Furthermore, it has also been documented in the literature that the different rates of film formation on the α and β phases can cause film fracture at α/β interfaces thereby initiating corrosion attack [82]. Moreover, based on the theory by Slámečka et al., the size, scatter, and depth of oxide granules on the implant surface may affect the fatigue resistance greatly by acting as crack initiation sites [84]. The formation and growth of a fatigue crack is more likely if the mechanical properties of the oxide layer and implant are of considerable discrepancy [85]. Therefore, the oxide layer can be either a defender or a security risk for internal fixation.

## 6. Allergic Reaction

Hypersensitivity to implanted metal device is considered as a type IV hypersensitivity, which is a delayed-type hypersensitivity (DTH) mediated by T lymphocytes [86]. It is agreed that metal hypersensitivity is mainly dependent on the following factors [86]: (1) Substance causing frank allergic reaction. For instance, nickel ions existing in the form of haptens can be identified by major histocompatibility complex (MHC) class II molecules in T-cell receptors; (2) Properties of metal materials. The release of metal ions first depends on the alloy components, surface modification, and chemical and physical corrosion factors; (3) Tissue environment. The sensitivity varies in different tissues and, of these, skin and subcutaneous tissue are the most sensitive; (4) Auxiliary factors. Infection is more likely to lead to allergic reactions; (5) Individual factors.

It has been estimated that failure rate of implantation in patients with metallic allergy is almost three times greater than that in the general population [87]. Cadosh et al. considered that osteoclast precursors can multiply and differentiate to mature osteoclast on the implant surface by establishing a corrosive model of stainless steel [88]. Hallab et al. justified the activation of T cells and B cells in metallic implant recipients [89]. This may in part reveal that implant-associated sensitizer is a very complicated immune response. This phenomenon was particularly evident in those suffering metal hypersensitivity [90]. Through observation, the mature osteoclasts were able to erode the implant and release metal debris into the surrounding tissue [91]. The wear out on the surface facilitates fatigue crack growth and infiltration of inflammatory cells inhibits the formation of oxide layer. Osteoporosis around the fixation greatly shortens the lifespan of implant because of osteoclast proliferation [92]. Debris can integrate with endogenous protein, thereby aggravating immune responses in turn [93]. However, the literature on the possible role of mechanism of metal sensitivity on fatigue crack growth is still non-existent.

Thyssen et al. explicitly indicated the diagnostic criteria for metal-allergy-associated dermatitis after orthopedic implantation: (1) Eczematous dermatitis in operation region confirmed as allergic rash by histological examination; (2) Positive patch test [94]. In this definition, patients with mild reaction merely feel occasional pruritus in the region of implantation without obvious systemic symptoms and influence of fracture healing. Push-through experiments have observed notable electrolysis on the internal fixation at a secondary procedure [95]. The implant is largely surrounded by recognizably dark grey and tough granulation tissue. Eczematous dermatitis can heal after removal of implant without the help of medication. By comparison, for severe reaction, the cutaneous area exposure to fixation takes on classical symptoms (infection-mimicking reactions), progressive dermatitis, burst, persistent effusion, formation of multiple fistula, and even systemic allergic reaction, which are less treatable. During the removal of the implant, some screws fastened through the conical thread holes have been seen to become loose, with a wealth of jelly-like necrotic tissues without bacterial growth in the secretion samples. Most of the manifestations can disappear immediately after necessitated removal. But chronic osteomyelitis and osteosclerosis may ensue in extreme cases treated improperly due in part to absence of blood supply.

The current study has verified that nickel ion is a potential sensitization factor [96]. With the passage of time, the quantity of nickel ions released from the surface of the metal increases in volume stepwise, thus they can evoke cells apoptosis, carcinogenicity and inflammation reaction [96]. Actually, the nickel content is approximately 10%~15% (mass fraction) in the most commonly used medical stainless steel-316 L. Given the possible harm from nickel-titanium (Ni-Ti) instruments, the best prevention is currently considered to limit the nickel content in stainless steel strictly. Cobalt-chromium-molybdenum (Co-Cr-Mo) alloys are another commonly used material characterized by a superior hardness and corrosion resistance. However, a local overload of cobalt ions also has direct cytotoxicity and a carcinogenic effect, and may even trigger a violent allergic reaction [97].

It has been reported that allergy to alloys is more common for pure metal, which may be related to an over-representation of the compound undesirable reaction caused by alloy [98]. The epidemiological characteristics of metal allergy have been altered in recent years. Incidence of allergy of cobalt alloys had fallen from 8.3% in 1997 to 0.8% in 2007. Yet it showed no significant change in the reported rate of nickel alloys allergy from 5.2% in 1977 to 4.6% in 2009 [99]. The incidence of positive patch test of both cobalt and nickel allergy decreased, while there was no pronounced change observed in chromium allergy. The aforementioned improvements are closely bound up with the optimization of fabrication technology. Furthermore, if a history of metal allergy is determined in the early healing period, the internal fixation should be removed in time [100].

## 7. New Trends

Traditional materials of internal fixation are mainly concerned with metal materials, which can promote the early fracture healing. However, the implantation is indeed traumatic for patient, and prone to induce stress shelter. A second procedure, without doubt, will impose suffering and economic burdens on patients. Recent trends in product offerings have been to accelerate the development of polymer materials (such as collagen, fiber, and polyamino acid) and absorbable inorganic materials (biodegradable ceramics) [24,101,102,103,104,105]. However, inevitably, when these artefacts reach the final stages of degradation, biodegradable fixation devices elicit a local foreign-body reaction [106]. Additionally, recent technology of shape memory alloy allows for prolonged lifespan of implant; this may represent a breakthrough in metal science [107]. Some studies have manifested significant benefits of shape memory alloy, distinguished by a remarkable shape memory effect, hyper-elasticity, biocompatibility, and resistance of corrosion and friction [108].

Micro-cracks are observed not only for reliable fixation of normal strength but also in the unloaded state [85]. As reflected in most studies, it is believed that the micro-cracks emerging before loading are basically responsible for the low tensile strength of the implant [21,109]. Some studies elucidated that the fatigue lifespan of additively manufactured lattice structures might make modest contributions. Material scientists manufactured the lattice structures made from Ti6Al4V with the combination of microstructural design and nano-scale surface engineering, like heat treatments, hot isostatic pressing (HIP), sand blasting, and chemical etching on the microstructure, which can markedly reduce the vulnerability and corrosion of lattice meta-biomaterials and inhibit fatigue crack initiation by increasing microstructural barrier with grain refinement [110]. Yang et al. adopted hydrothermal technology on surface mechanical attrition treated (SMATed) titanium (S-Ti) to fabricate a TiO2 nanorods (TNR)-arrayed coating [111]. The surface of Ti-based implants was coated with a layer of TNR/S-Ti, thus improving corrosion fatigue, osteogenesis, and fatigue lifespan [112]. Javier Trinidad found that the mechanical property of porous magnesium achieved the desired consistency with trabecular and cortical bone when the porosity varied from 30% to 69% [113]. As an added benefit, magnesium can form a soluble and non-toxic oxide in vivo, which was excreted in urine [114].

Assuming that the innovations mentioned above can be implemented, the ideal rehabilitation would be based on the absorption rate of internal fixation to adjust the level of activity, which would avoid stress shelter effect and maintain a balance between absorption of implant and bone reconstruction. It also would promote the early fracture healing as a result of releasing growth factors. Despite the gradual degradation, the strength of plate would not decrease at various stages of loading, and stress concentrations would be transferred to the remodeling bone. This scenario has great scientific potential and practical significance. For non-absorbable fixation, the implant should be removed after fracture healing in principle, unless the patient is elderly and in poor health.

## 8. Conclusions

Given the lack of specific understanding and consensus, previous investigations on fatigue crack propagation in internal fixation were relatively scattered and imperfect. The current experimental conditions are unable to mimic exactly what happens in the human body to implant due to the great internal complexity. The stress conditions and internal environment of implant vary from place to place in vivo. Therefore, at present, how to perfect a fixation device remains a major challenge for fixation design and processing technology. In this review, the possible causes of fatigue crack growth in actual application of implant were elaborated scientifically from the perspectives of underlying crack propagation mechanisms, inappropriate use, design flaws, elastic modulus, biocompatibility, corrosion resistance, and metal allergy. In this way, clinicians, manufacturers, and researchers can better understand the future development of internal fixation devices, and acquire a relevant theoretical basis for further studies on fatigue crack growth.

## Figures and Tables

**Figure 1 materials-14-00176-f001:**
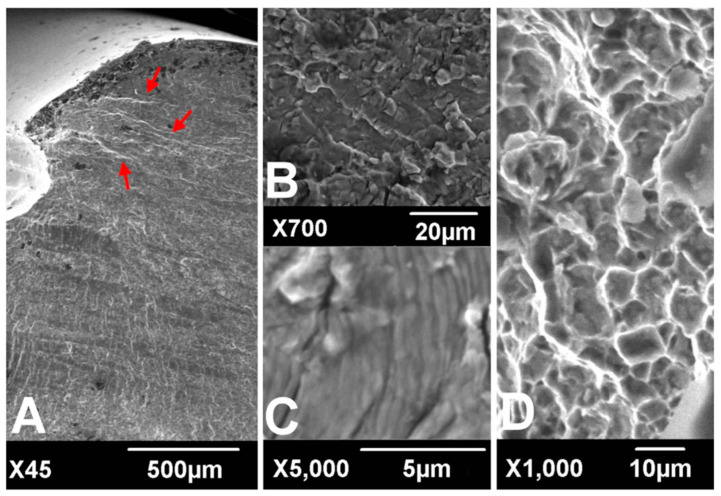
Scanning electronic microscopic findings of fracture surfaces. River marks (red arrows) originate from the thread crest (**A**). Transgranular cracking (**B**) with striations perpendicular to the river marks (**C**) was observed at the crack propagation zone. Dimpled ductile fractures were observed at the overload zone (**D**). (according to [6]).

**Figure 2 materials-14-00176-f002:**
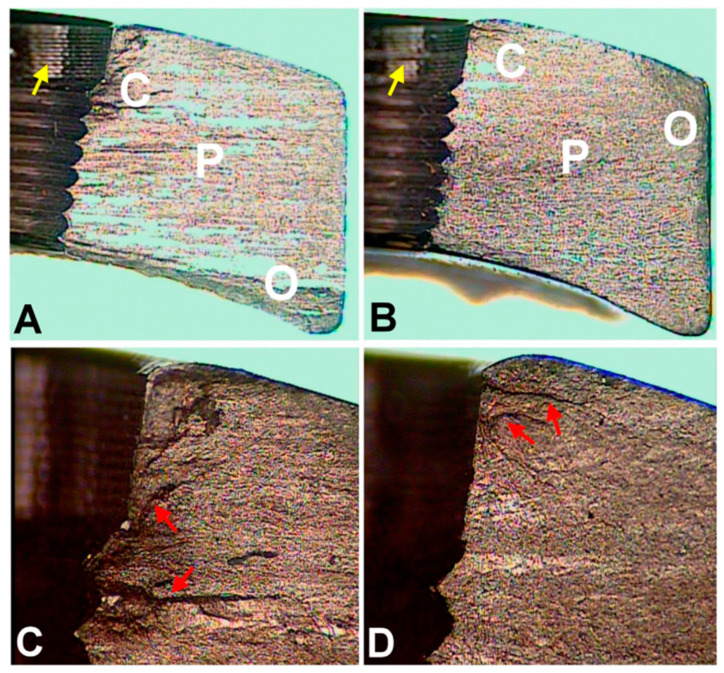
Optical microscopic images of fracture surfaces. There are three zones indicating fatigue fracture (**A**,**B**) and the magnified views of the crack initiation sites (**C**,**D**). C represents the crack initiation zone. P represents the crack propagation zone. O represents the final overloading zone. Red arrows point to the crack lines. Yellow arrows point to the machining lines at the screw holes. (according to [6]).

**Figure 3 materials-14-00176-f003:**
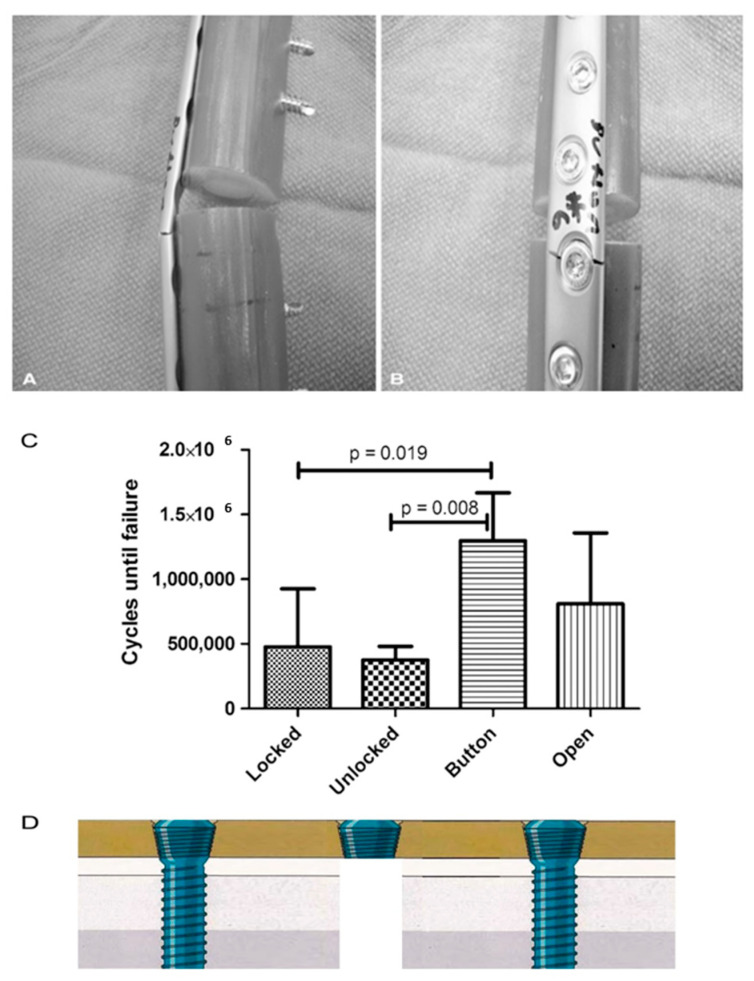
(**A**,**B**) All plates failed through a screw hole adjacent to the defect as is seen on the (**A**) left and (**B**) right in these representative Button samples. (**C**) The fatigue life in cycles to failure is shown for each plate and screw configuration. (according to [46]). (**D**) The authors did not to date test the strength of the construct near the fracture site for using a single button.

**Figure 4 materials-14-00176-f004:**
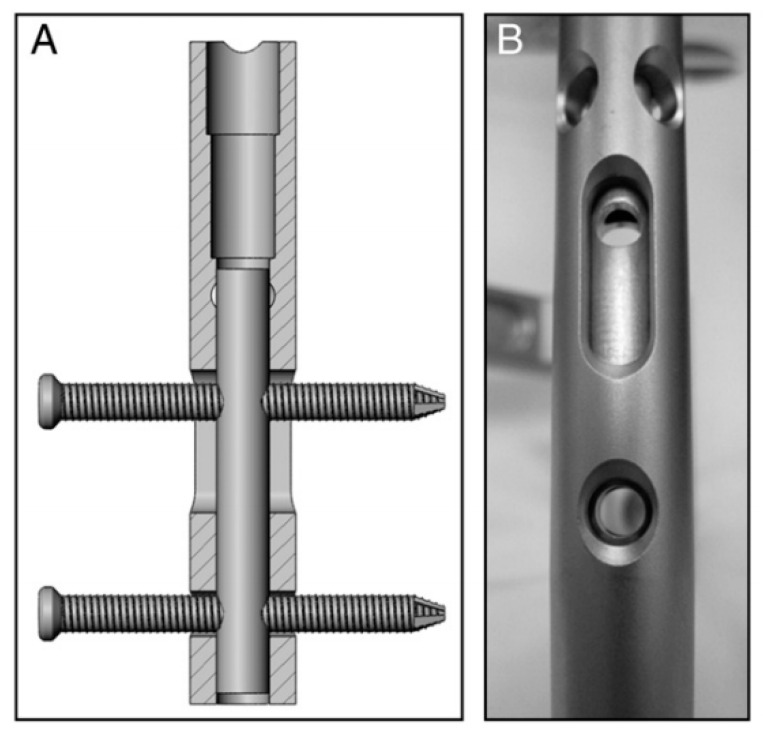
(**A**,**B**) Prior to implantation of the nails, a micromotion insert was placed in the proximal cannulus of each nail. An insert in the proximal nail stem was used to align two 3.9-mm locking bolts—one in the dynamic slot and one in the 5-mm static locking hole. This configuration produced 1.1 mm of free axial travel (according to [54]).

**Figure 5 materials-14-00176-f005:**
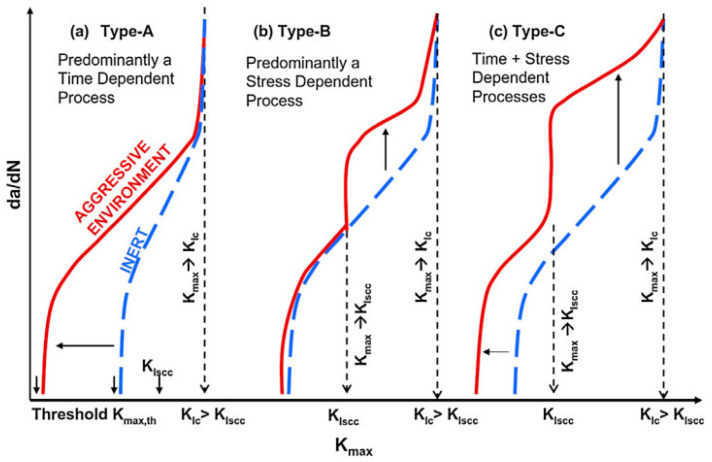
Schematic representation of crack growth under different scenarios of simultaneous cyclic and tensile loading in inert and corrosive environment (according to [75]).

**Figure 6 materials-14-00176-f006:**
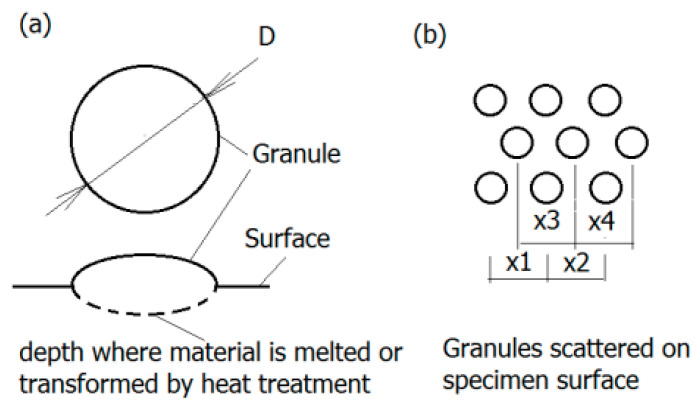
Granular surface: (**a**) Geometry of granules—The region of melted and transformed material is marked dashed line; (**b**) granules displacement on the surface (according to [78]).

## Data Availability

Data sharing not applicable. No new data were created or analyzed in this study. Data sharing is not applicable to this article.

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
