# Peer review of "Fatigue Crack Growth and Fracture of Internal Fixation Materials in In Vivo Environments—A Review"

_materials, 2021, doi:10.3390/ma14010176_

Round 1
Reviewer 1 Report
This article is written at a high level, is comprehensible, readable and complies with the relevant standards for its processing. The procedure for processing the experimental part was chosen correctly, it has a logical arrangement and good photo documentation. The obtained results are interesting and can be used in further research.
I have no serious formal or substantive comments on this article.
Author Response
Reviewer #1:
This article is written at a high level, is comprehensible, readable and complies with the relevant standards for its processing. The procedure for processing the experimental part was chosen correctly, it has a logical arrangement and good photo documentation. The obtained results are interesting and can be used in further research.
I have no serious formal or substantive comments on this article.
Response: Thank you very much for confirming the contribution of our current work in the field.
Reviewer 2 Report
The reviewed paper is an interesting review on fatigue and fracture of internal fixation materials. The authors have their rich achievements in this area, but they do not use in this work. Therefore, I have a few small comments that must be taken into account.
- Please, if possible, change all the figures that have been copied from other works.
- There is no assessment of the strengths and weaknesses of specific solutions. They don't have to be quantified (because they don't have their own results), but this is missing from the conclusions.
- Please add some citations, about crack investigations and resistant to failure and control of this phenomenon, in the Introduction section:
- Macek, W.; Branco, R.; Szala, M.; Marciniak, Z.; Ulewicz, R.; Sczygiol, N.; Kardasz, P. Profile and Areal Surface Parameters for Fatigue Fracture Characterisation. Materials 2020, 13, 3691. https://doi.org/10.3390/ma13173691 or
- W. Macek, R. Branco, J. Trembacz, J.D. Costa, J.A.M. Ferreira, C. Capela, Effect of multiaxial bending-torsion loading on fracture surface parameters in high-strength steels processed by conventional and additive manufacturing, Engineering Failure Analysis, Volume 118, 2020, 104784, ISSN 1350-6307, https://doi.org/10.1016/j.engfailanal.2020.104784.
Author Response
The reviewed paper is an interesting review on fatigue and fracture of internal fixation materials. The authors have their rich achievements in this area, but they do not use in this work. Therefore, I have a few small comments that must be taken into account.
Point 1.Please, if possible, change all the figures that have been copied from other works.
Response 1: Thank you for your good suggestions. We provide a summary and a synthesis of the findings of selected high-quality research contributions being published by other authors. Despite the various levels of complexity of the technical topics, the article is limited in including analytical and experimental parts from cited papers. We also seek a well balance between the figures and text so that the addressed issues are clearly stated in a simple and efficient way.
Therefore, no changes have been made to the cited figures.
Point 2.There is no assessment of the strengths and weaknesses of specific solutions. They don't have to be quantified (because they don't have their own results), but this is missing from the conclusions.
Response 2: In our review, the possible causes of fatigue crack growth in actual application of implant have been well elaborated. We hope that clinicians, manufacturers and researchers can better understand the future development of internal fixation devices, and acquire a relevant theoretical basis for further studies on fatigue crack growth. However, this article does not include the detailed solutions. In addition, our review cannot indicate the quality of specific solutions in the absence of controlled study or Meta-analysis. We have not made any further changes in the conclusion but will add additional data in the future research paper. Thank you.
Point 3. Please add some citations, about crack investigations and resistant to failure and control of this phenomenon, in the Introduction section:
- Macek, W.; Branco, R.; Szala, M.; Marciniak, Z.; Ulewicz, R.; Sczygiol, N.; Kardasz, P. Profile and Areal Surface Parameters for Fatigue Fracture Characterisation. Materials 2020, 13, 3691. https://doi.org/10.3390/ma13173691 or
- W. Macek, R. Branco, J. Trembacz, J.D. Costa, J.A.M. Ferreira, C. Capela, Effect of multiaxial bending-torsion loading on fracture surface parameters in high-strength steels processed by conventional and additive manufacturing, Engineering Failure Analysis, Volume 118, 2020, 104784, ISSN 1350-6307, https://doi.org/10.1016/j.engfailanal.2020.104784.
Response 3: We have added these two articles to our ref [12, 13]. Thank you.
Reviewer 3 Report
I should say something about my background.
I am a native English speaker
I know a fair amount about fracture mechanics.
I just had a hip replacement!
But I am not a specialist in creep fracture of implants, nor in medicine.
So I would simply like to report that I read the paper with interest, and to focus on it I edited the English. I cannot promise I always preserved the sense of it, but I tried. My recommendation is to publish after the English has been cleared up, and I hope that the attachment assists with that. Since I used change tracking in Word, I am no longer anonymous, but I am not going to worry about that.

Author Response
I should say something about my background.
I am a native English speaker
I know a fair amount about fracture mechanics.
I just had a hip replacement!
But I am not a specialist in creep fracture of implants, nor in medicine.
So I would simply like to report that I read the paper with interest, and to focus on it I edited the English. I cannot promise I always preserved the sense of it, but I tried. My recommendation is to publish after the English has been cleared up, and I hope that the attachment assists with that. Since I used change tracking in Word, I am no longer anonymous, but I am not going to worry about that.
Response: We really appreciate your precious time and great efforts on reviewing the manuscript. We have carefully addressed all the review comments and improved the quality of this manuscript accordingly. We would like to extend our heartfelt thanks to you!